

# Comparative analysis of *MAPK* and *MKK* gene families reveals differential evolutionary patterns in *Brachypodium distachyon* inbred lines

Min Jiang[1,2,*], Peng Li[1,*] and Wei Wang[1]

[1] Shanghai Key Laboratory of Plant Functional Genomics and Resources, Shanghai Chenshan Plant Science Research Center, Shanghai Institutes for Biological Sciences, Chinese Academy of Sciences (CAS), Shanghai Chenshan Botanical Garden, Shanghai, China
[2] Ministry of Education Key Laboratory for Biodiversity Science and Ecological Engineering, Institute of Biodiversity Science, School of Life Sciences, Fudan University, Shanghai, China
* These authors contributed equally to this work.

Corresponding author
Min Jiang, yijinsha@126.com

## ABSTRACT

**Background:** Mitogen-activated protein kinase (MAPK) cascades are involved with signal transduction in almost every aspect of plant growth and development, as well as biotic and abiotic stress responses. The evolutionary analysis of MAPKs and MKKs in individual or entire plant species has been reported, but the evolutionary patterns in the diverse inbred lines of *Brachypodium distachyon* are still unclear.
**Results:** We conducted the systematical molecular evolutionary analysis of *B. distachyon*. A total of 799 MAPKs and 618 MKKs were identified from 53 *B. distachyon* inbred lines. Remarkably, only three inbred lines had 16 MPKs and most of those inbred lines lacked MPK7-2 members, whereas 12 MKKs existed in almost all *B. distachyon* inbred lines. Phylogenetic analysis indicated that MAPKs and MKKs were divided into four groups as previously reported, grouping them in the same branch as corresponding members. MPK21-2 was the exception and fell into two groups, which may be due to their exon-intron patterns, especially the untranslated regions (UTRs). We also found that differential evolution patterns of MKK10 paralogues from ancient tandem duplicates may have undergone functional divergence. Expression analyses suggested that *MAPKs* and *MKKs* likely played different roles in different genetic contexts within various tissues and with abiotic stresses.
**Conclusion:** Our study revealed that UTRs affected the structure and evolution of *MPK21-2* genes and the differential evolution of *MKK10* paralogues with ancient tandem duplication might have functional divergences. Our findings provide new insights into the functional evolution of genes in closely inbred lines.

## INTRODUCTION

Mitogen-activated protein kinase (MAPK or MPK) signaling cascades play vital roles in the stress response, cell division, and developmental regulation. They are divided into three

highly-conserved subfamilies that continuously act in a sequential manner in evolution and fundamental signaling transduction pathways (*Rodriguez, Petersen & Mundy, 2010*; *Xu & Zhang, 2015*; *Jagodzik et al., 2018*). The MAPK kinase kinases (MKKKs or MEKKs) are activated by extracellular cues and subsequently phosphorylate and activate the S/T-X$_{3-5}$-S/T motif of downstream MAPK kinases (MAPKKs or MKKs), which, in turn, phosphorylate and activate MAPKs at their TXY activation loop (*Rodriguez, Petersen & Mundy, 2010*; *Singh et al., 2012*). Activated MPKs regulate downstream cellular targets, including regulatory and metabolic enzymes and transcription regulators (*Joo et al., 2008*, *Guan et al., 2014*).

*Brachypodium distachyon* ($2n = 10$) is an annual temperate grass with a close phylogenetic relationship to other temperate cereals and an intermediate position within the Pooideae subfamily (*Soreng et al., 2015*; *Catalan et al., 2016*). *B. distachyon* is desirable for its small physical stature, rapid life cycle, ability to self-fertilize, and small diploid genome size (*Draper et al., 2001*; *Garvin et al., 2008*). Highly efficient *Agrobacterium*-mediated transformation methods in Brachypodium have also been established (*Vain et al., 2008*; *Vogel & Hill, 2008*). Therefore, *B. distachyon* is widely used as a model plant for studying problems unique to cereals and grasses (*Vogel et al., 2010*; *Brkljacic et al., 2011*; *Mur et al., 2011*; *Catalan et al., 2014*). The morphological, molecular, and cytological analyses of diverse *B. distachyon* inbred lines have been conducted (*Filiz et al., 2009*; *Vogel et al., 2009*) and their nuclear and plastid genomes have been deep sequenced and annotated (*Gordon et al., 2014*; *Gordon et al., 2017*; *Sancho et al., 2018*). Further analysis showed that the inbred lines of *B. distachyon* are divided into three different genomic groups, including a mostly Extremely Delayed Flowering (EDF+) clade, a mostly Spanish (S+) clade, and a Turkish (T+) clade, based on their flowering phenotype and geographical substructure (*Sancho et al., 2018*).

To date, the evolutionary mechanisms of MAPK cascades in plants have indicated a diverse domain organization and novel activation loop variants (*Mohanta et al., 2015*) and/or distinct expansion mechanism (*Jiang & Chu, 2018*). A variety of single-gene duplication types emerge continuously and have involved in the plant's adaptation to dramatically changing environments (*Wang et al., 2009*; *Cuevas et al., 2016*). However, whole-genome duplications (WGDs) are considered to be a major force in the evolution of morphological and physiological diversity (*Soltis et al., 2009*; *Paterson et al., 2010*). The ancient tandem duplication event occurred at the adjacent genes in the same chromosome, which are usually expanded or retained by an unequal crossing (*Freeling, 2009*). Tandem duplication often displays less expression difference and functional divergence than distant duplication (*Makino & McLysaght, 2012*; *Ghanbarian & Hurst, 2015*). However, there is limited information on the gene expansion mechanism and functional evolution of the MAPK cascades in diverse *B. distachyon* inbred lines, including for Bd21 (*Chen et al., 2012*; *Jiang et al., 2015*). We studied the evolutionary patterns of MAPKs and MKKs from different *B. distachyon* inbred lines. The phylogenetic relationships and the identification of MAPKs and MKKs were determined for 53 *B. distachyon* inbred lines. We investigated gene and domain construction patterns of the

individual members with a certain divergence and focused on the evolutionary history of *MKK10* paralogues in different *B. distachyon* inbred lines. This revealed various conservative and divergent tandem gene clusters. The expression patterns of these genes were analyzed in Bd21, BdTR8i, and Bd30-1 from three genetic groups in various tissues and abiotic stresses, and their potential functions were also investigated.

## METHODS AND MATERIALS

### Identification of *MAPK and MKK* gene family members

We downloaded gene information for *MAPK* and *MKK* from *B. distachyon* Bd21 from the PLAZA platform (https://bioinformatics.psb.ugent.be/plaza/) (*Van Bel et al., 2018*). BLASTP (*Altschul et al., 1997*) searches were conducted with a threshold of 90% identity; searches were performed with orthologous protein sequences using BdMAPKs and BdMKKs as queries in *BrachyPan* (https://brachypan.jgi.doe.gov/) (*Gordon et al., 2017*) to identify these genes in the 53 diverse *B. distachyon* inbred lines. Collected sequences were only accepted for scanning using InterPro software (*Mitchell et al., 2019*) if they harbored MAPK or MKK consensus sequences, including the activation loop TXY motif for MPKs, the active site motif D(L/I/V)K, and the phosphorylation target site S/T-$X_5$-S/T within the activation loop for MKKs. The gene identifier information of these sequences was collected and is listed in Tables S1 and S2.

### Gene structure and sequence alignments

The exon/intron structure of identified *MAPKs* and *MKKs* was performed using Gene Structure Display Server 2.0 (GSDS 2.0) software (http://gsds.gao-lab.org/). All of the full-length amino acid sequences were initially aligned using Clustal Omega with default parameters (http://www.ebi.ac.uk/Tools/msa/clustalo/). The domains and motifs of MAPKs and MKKs were scanned using InterProScan software (http://www.ebi.ac.uk/interpro/) (*Jones et al., 2014*). The structural schematic of all members of MAPK and MKK were executed according to InterProScan analysis results. The alignment logos of the protein conserved domain were generated using the WebLogo3 application (http://weblogo.threeplusone.com/).

### Synteny and phylogenetic analyses

The phylogenetic relationships of all 53 *B. distachyon* inbred lines were generated in the *BrachyPan* project and visualized with the CorelDRAW X3 program. Phylogenetic trees were created based on the alignment of all MAPKs or MKKs using the maximum likelihood (ML) method with the Jones–Taylor–Thornton (JTT) model, 2,000 bootstrap values, and partial deletion by the MEGA 6.0 software, respectively (*Tamura et al., 2013*). The Neighbor Joining (NJ) Trees of the MAPKs or MKKs were also reconstructed with the same parameters using MEGA 6.0. We obtained the synteny information of duplicate genes and the tandem (TD) data from the PlantDGD database (http://pdgd.njau.edu.cn:8080/) (*Qiao et al., 2019*).

## Plant sample preparation

We sowed BdTR8i, Bd21, and Bd30-1 seeds in ½ MS medium in the dark for 4 d at 25 °C and then transferred them to a soil mix. Plants were grown in a greenhouse under 14 h light (21 °C)/10 h dark (18 °C) photoperiods. We harvested the root, stem, leaf blade, and leaf sheath at the eight-to-nine leaf stage. Spikelet samples from *B. distachyon* were collected at the early flowering stages according to their different flowering times (Fig. S1). For the abiotic stress treatment, 2-week-old *B. distachyon* seedlings were dipped in ½ MS liquid medium containing 20% PEG 6000 and 200 mM NaCl, and then plants were collected after treatment for 3 h and 6 h, respectively. Moreover, seedlings were transferred to a growth chamber and heat-treated at 40 °C for 3 h and 6 h. All samples were flash frozen in liquid nitrogen and stored at −80 °C for RNA extraction.

## Expression analysis

Total RNA was extracted from samples using Trizol reagent and 1–2 µg was reverse-transcribed into cDNA using PrimeScript RT Master Mix Perfect Real Time (TaKaRa, Beijing, China) according to the manufacturer's instructions. The quality of total RNA was detected using Nanodrop1000 and its integrity was estimated by electrophoresis in 1.5% (w/v) agarose gel. The real-time quantitative polymerase chain reaction (RT-qPCR) was carried out in 10 µl reactions with 5–50 ng of first-stand cDNA products (four µl), five pmol of each primer (0.4 µl), five µl SYBR green master mix (2X), 0.2 µl ROX as a passive reference standard to normalize the SYBR fluorescent signal. The conditions for RT-qPCR were: initial activation at 95 °C for 5 min followed by 45 cycles of 95 °C for 30 s, and 60 °C for 30 s. Subsequently, the specificity of PCR products was monitored using a melting curve analysis (61–95 °C with fluorescence read every 0.5 °C). The *B. distachyon* actin (gene locus: *Bradi2g24070*) gene was used as an internal control for all RT-qPCR analyses; specific primers for *MAPK* and *MKK* were listed in Table S3. Three independent biological replicates were conducted for each experiment. The relative expression of *MAPK* and *MKK* genes was calculated using the $2^{-\Delta\Delta Ct}$ method.

# RESULTS

## Identification and annotation of MPKs and MKKs in 53 diverse *B. distachyon* inbred lines

We identified the two gene families by searching homologous Bd21 sequences in the public *BrachyPan* database (Chen et al., 2012) to determine the conservation and divergence of MPKs and MKKs in 53 diverse *B. distachyon* inbred lines. All predicted MPKs and MKKs were named based on the similarity of their orthologous protein to that of *A. thaliana* and *B. distachyon* (Ichimura et al., 2002; Chen et al., 2012). Ultimately, a total of 799 MPKs and 618 MKKs were retrieved (Table 1; Tables S4 and S5). We found that most *B. distachyon* inbred lines had 14 or 15 MPKs apart from Bd21, BdTR3c, and Bd18-1, which had 16 members (Table 1). Further analysis showed that only seven inbred lines had the *MPK7-2* gene, including Bd21, Bd2-3, BdTR3c, Bd18-1, S8iiC, Mur1, and Foz1 (Table S1). This may be the result of an incomplete annotation of the genome sequence or the long sequence of the MPK7-2 protein, which usually consists of 1,708

amino acid (aa) residues. Most *B. distachyon* inbred lines harbored 12 MKK members except Tek-4 (11), Bd3-1 (7), Adi-10 (10), Gaz-8 (5), ABR5 (11), Foz1 (11), and Jer1 (11) (Table 1; Table S5). The incomplete assembly of Tek-4 (77.82%), Bd3-1 (89.52%), Adi-10 (89.52%), and Gaz-8 (88.31%) may be the reason that relatively few MKK members have been identified (*Gordon et al., 2017*). Further analysis showed that the *B. distachyon* inbred lines lacked a particular MKK member; for example, MKK10-5 of Tek-4, MKK10-4 of ABR5, MKK4 of Foz1, MKK5 of Jer1, MKK10-3 and -4 of Adi-10 (Table S2). We also incorporated the available genomic detailed information from MPKs and MKKs (Tables S1 and S2).

## Phylogenetic classification of *B. distachyon* inbred lines MAPKs and MKKs

To investigate the phylogenetic relationship of MPK proteins in diverse *B. distachyon* inbred lines, the phylogeny of all identified 799 MPK protein sequences were performed using ML and NJ methods, respectively. As expected, all homologues for each of the 16 Bd21 MPKs (BdMPKs) were divided into four groups (A, B, C, and D) and clustered on the corresponding branch except Tek-4MPK16 (Fig. 1; Figs. S2 and S3). Tek-4MPK16 consisted of only 183 aa, while the other MPK16 members had 544 aa (Table S1). MPK21-2 had two branches, designated as type I and II (Table S6), indicating that it may have a certain functional divergence. In addition, MPK7-1 and MPK7-2 were located on same branch with a large discrepancy in their lengths (Fig. 1), suggesting functional divergence, which is supported by previous functional studies (*Jiang et al., 2015*).

We analyzed a total of 618 MKKs for their phylogenetic relationship with corresponding protein sequences using ML and NJ methods, respectively. Almost all of the orthologous genes for each of the 12 Bd21 MKKs (BdMKKs) had similar clustering patterns with corresponding branches and fell into four groups: A, B, C, and D (Fig. 2; Figs. S4 and S5). However, BdTR10cMKK10-3, Jer1MKK10-4, Mur1MKK10-5, and BdTR13cMKK10-5 were branched out from the other members, suggesting that these gene members may have diverged (Fig. 2). It is noteworthy that Mur1MKK10-5 (188 aa) and BdTR13cMKK10-5 (208 aa) had shorter amino acids than other members, which usually contained 332 aa (Table S2). BdTR10cMKK10-3 (163 aa) was also shorter relative to the normal MKK10-3 (344 aa). In contrast, Jer1MKK10-4 (424 aa) was longer when compared with the typical MKK10-4 (341 aa) (Table S2). These situations may affect their evolutionary relationship with MKKs from other *B. distachyon* inbred lines.

## Exon-intron compositions and length variations of MPKs and MKKs in *B. distachyon* inbred lines

The abundance of non-protein-coding DNA within a genome, such as an intron, increased consistently with the genome complexity (*Taft, Pheasant & Mattick, 2007*). Intron pattern analyses can enhance our understanding of the structure and evolution of genes (*Zhang et al., 2014*). We also surveyed the exon-intron architecture of different MPKs and MKKs using GSDS software to elucidate the relationship or divergence among paralogues and orthologues. Most members showed similar exon-intron patterns with the intron

**Table 1 Number of *B. distachyon* inbred lines MPKs and MKKs identified genes from the *BrachyPan* database and their associated information.**

| Genetic groups | Inbred line | Latitude (*Gordon et al., 2017*) | Longitude | Elevation (m) | Ploidy | MPKs | MKKs |
|---|---|---|---|---|---|---|---|
| EDF+ | Arn1 | 42° 15′ 23.44″ N | 0° 43′ 47.46″ E | 681 | – | 15 | 12 |
| | Mon3 | 41° 39′ 4.75″ N | 0° 12′ 37.51″ W | 515 | diploid | 15 | 12 |
| | Bd1-1 | 39° 11′ 27.44″ N | 27° 36′ 28.59″ E | 141 | diploid | 14 | 12 |
| | ABR9 | – | – | – | – | 15 | 12 |
| | Bd29-1 | 44° 30′ 55″ N | 33° 33′ 23″ E | 260 | diploid | 15 | 12 |
| | Tek-4 | 41° 0′ 40.1″ N | 27° 31′ 8.8″ E | 20 | diploid | 14 | 11 |
| | BdTR7a | 39° 44′ 53.45″ N | 34° 39′ 1.15″ E | 1,035 | diploid | 15 | 12 |
| | Tek-2 | 41° 0′ 40.1″ N | 27° 31′ 8.8″ E | 20 | diploid | 15 | 12 |
| | BdTR8i | 37° 6′ 31.87″ N | 34° 4′ 17.06″ E | 2,385 | diploid | 15 | 12 |
| T+ | Bd21 | 33° 45′ 39.18″ N | 44° 24′ 11.07″ E | 42 | diploid | 16 | 12 |
| | Bd21-3 | 33° 45′ 39.19″ N | 44° 24′ 11.08″ E | 43 | diploid | 14 | 12 |
| | Bd3-1 | 33° 45′ 39.19″ N | 44° 24′ 11.08″ E | 43 | diploid | 14 | 7 |
| | Bd2-3 | 33° 45′ 39.18″ N | 44° 24′ 11.07″ E | 42 | diploid | 14 | 12 |
| | Adi-10 | 37° 46′ 14.5″ N | 38° 21′ 8.2″ E | 510 | diploid | 15 | 10 |
| | BdTR12c | 39° 44′ 53.45″ N | 34° 39′ 1.15″ E | 1,035 | diploid | 15 | 12 |
| | Adi-2 | 37° 46′ 14.5″ N | 38° 21′ 8.2″ E | 510 | diploid | 15 | 12 |
| | Adi-12 | 37° 46′ 14.5″ N | 38° 21′ 8.2″ E | 510 | diploid | 15 | 12 |
| | BdTR9k | 39° 45′ 10.62″ N | 30° 47′ 19.07″ E | 932 | diploid | 15 | 12 |
| | Kah-1 | 37° 44′ 2.3″ N | 38° 32′ 0.2″ E | 665 | diploid | 14 | 12 |
| | Kah-5 | 37° 44′ 2.3″ N | 38° 32′ 0.2″ E | 665 | diploid | 15 | 12 |
| | BdTR5i | 40° 23′ 37.13″ N | 32° 59′ 7.32″ E | 1,596 | diploid | 15 | 12 |
| | BdTR10c | 37° 46′ 41.64″ N | 31° 53′ 5.68″ E | 1,288 | diploid | 15 | 12 |
| | BdTR11a | 38° 25′ 0.42″ N | 28° 1′ 52.75″ E | 986 | diploid | 14 | 12 |
| | BdTR11i | 39° 44′ 17.39″ N | 28° 2′ 24.71″ E | 363 | diploid | 15 | 12 |
| | BdTR11g | 41° 25′ 17.86″ N | 27° 28′ 36.81″ E | 124 | diploid | 15 | 12 |
| | BdTR13c | 39° 24′ 46.28″ N | 32° 59′ 17.24″ E | 1,192 | diploid | 15 | 12 |
| | BdTR13a | 39° 45′ 23.35″ N | 32° 25′ 56.46″ E | 787 | diploid | 15 | 12 |
| | Bis-1 | 37° 52′ 35.6″ N | 41° 0′ 54.3″ E | 529 | diploid | 15 | 12 |
| | Koz-3 | 38° 9′ 8.2.6″ N | 41° 36′ 34.8″ E | 853 | diploid | 14 | 12 |
| | Koz-1 | 38° 9′ 8.2.6″ N | 41° 36′ 34.8″ E | 853 | diploid | 15 | 12 |
| | BdTR3c | 36° 46′ 58.92″ N | 32° 57′ 46.71″ E | 1,957 | diploid | 16 | 12 |
| | Gaz-8 | 37° 7′ 39.8″ N | 37° 23′ 26.9″ E | 891 | diploid | 15 | 5 |
| | BdTR1i | 38° 5′ 35.03″ N | 28° 34′ 59.02″ E | 841 | diploid | 15 | 12 |
| | BdTR2b | 40° 4′ 55.55″ N | 31° 19′ 52.01″ E | 667 | diploid | 15 | 12 |
| | BdTR2g | 40° 23′ 37.13″ N | 32° 59′ 7.32″ E | 1,596 | diploid | 15 | 12 |
| | Bd18-1 | 39° 22′ 4.25″ N | 33° 43′ 48.91″ E | 1,101 | diploid | 16 | 12 |
| S+ | Bd30-1 | 36° 59′ 25.76″ N | 3° 33′ 31.44″ W | 1,220 | diploid | 15 | 12 |
| | ABR5 | 42° 34′ 23.45″ N | 0° 33′ 49.39″ W | 828 | diploid | 15 | 11 |
| | Mig3 | 42° 8′ 52.76″ N | 0° 11′ 41.89″ W | 572 | diploid | 15 | 12 |
| | Uni2 | 42° 7′ 3.98″ N | 0° 26′ 42.81″ W | 480 | diploid | 15 | 12 |
| | Mur1 | 42° 06′ 18″ N | 0° 51′ 23″ E | 487 | diploid | 14 | 12 |

| Table 1 (continued) | | | | | | | |
|---|---|---|---|---|---|---|---|
| Genetic groups | Inbred line | Latitude (*Gordon et al., 2017*) | Longitude | Elevation (m) | Ploidy | MPKs | MKKs |
| | Foz1 | 42° 38′ 11.44″ N | 1° 18′ 17.42″ W | 434 | diploid | 15 | 11 |
| | ABR2 | 43° 36′ 15.343″ N | 3° 15′ 46.580″ E | 371 | diploid | 14 | 12 |
| | ABR3 | 42° 10′ 49.8″ N | 0° 4′ 23.2″ W | 1,928 | diploid | 14 | 12 |
| | ABR4 | 42° 15′ 45.54″ N | 0° 43′ 0.48″ E | 480 | – | 15 | 12 |
| | ABR6 | 42° 34′ 27.48″ N | 2° 11′ 5.39″ W | 484 | – | 15 | 12 |
| | ABR7 | 41° 35′ 23.86″ N | 4° 45′ 24.26″ W | 725 | – | 14 | 12 |
| | S8iiC | 41° 36′ 19.3″ N | 0° 08′ 38.4″ E | 144 | – | 15 | 12 |
| | Jer1 | 42° 3′ 16.56″ N | 0° 0′ 44.57″ W | 418 | – | 14 | 11 |
| | Per1 | 42° 44′ 13.34″ N | 1° 44′ 58.6″ W | 742 | – | 14 | 12 |
| | Luc1 | 42° 36′ 36.18″ N | 0° 53′ 35.48″ W | 597 | – | 15 | 12 |
| | RON2 | 42° 46′ 50″ N | 0° 57′ 48″ W | 594 | – | 15 | 12 |
| | Sig2 | 42° 36′ 46.55″ N | 1° 0′ 52.38″ W | 524 | – | 14 | 12 |

number, exon length, and intron phase. The intron number was found to be relatively constant in three genetic groups, with the exceptions of MPK20-3, MPK21-2, and MKK3-3 (Fig. 3). Remarkably, the number of introns from MPK11, MPK21-1, and MKK1 in all *B. distachyon* inbred lines consistently contained 5, 10, and 8 introns, respectively (Fig. 3). Almost all MPK20-3 had three introns in group T+ apart from BdMPK20-3, Bd21-3MPK20-3, BdTR13aMPK20-3, BdTR3cMPK20-3, and Bd18-1MPK20-3 which had eight introns, while other two genetic groups of MPK20-3 displayed three or eight introns (Fig. 3A; Fig. S6). Moreover, the number of introns of MKK3-3 was highly consistent in group S+, and T+ usually contained nine introns. In contrast, the intron numbers in group EDF+ were highly variable; for example, ABR9MKK3-3, Bd29-1MKK3-3, Tek-4MKK3-3 had eight, six, and four introns, respectively (Fig. 3B; Fig. S7).

The exon-intron patterns of MPK21-2 fell into two groups, which coincided with their phylogenetic relationship mentioned above (Figs. 1 and 4). However, the phylogenetic relationship was not completely consistent between the reconstructed full length coding sequence (CDS) and their exon-intron patterns (Fig. 4). Further analysis showed that all type II MPK21-2s harbored UTRs, while type I MPK21-2s had no UTR except for RON2MPK21-2 and Tek-4MPK21-2 (Fig. 4). Most MPK21-2s had seven introns in group EDF+ and S+ or nine introns in group T+ except for BdMPK21-2, Bd3-1MPK21-2, and Adi-10MPK21-2 (Fig. 3A). These results suggest that the structure and evolution of these genes were influenced by intron patterns and may be affected by UTRs.

## Common conserved domain analysis of *B. distachyon* MPKs and MKKs

Previous research has reported that MAPKs had several conserved domains or signature sequences with vital structural or functional roles, including the GxGxxG motif in the nucleotide binding (NB) domain (*Mohanta et al., 2015*), the TXY motif in the activation loop (*Xu & Zhang, 2015*), D(I/L/V)K motif in the active site (*Goyal et al., 2018*), and the common docking (CD) domain in the C-terminal extension region outside the

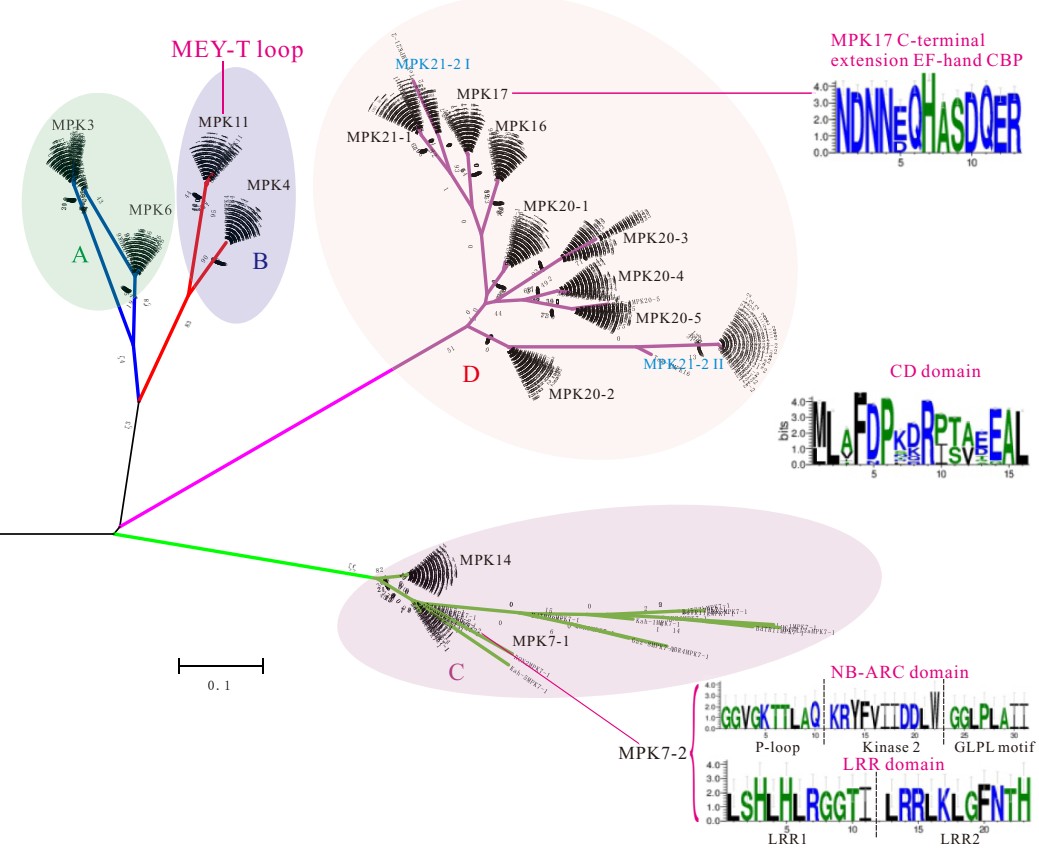

**Figure 1** **Phylogenetic distribution of 799 MPKs among diverse *B. distachyon* inbred lines.** MPKs are divided into four clades (A–D). Unique to *B. distachyon* MPK17s, an EF-hand CBP domain is found in their C-terminal extensions. The group A, B and C carry TEY T-loops and group D MPKs carry TDY motifs, with the exception of MPK11s which carry an MEY motif. The CD domain is conserved in the MAPK family. The negatively charged amino acids (D and E) are expected to be exposed to the surface of the molecules. MPK7-2 also contained NB-ARC domain and LRR domain. NB: nucleotide-binding; LRR: leucine-rich repeat. ARC: APAF-1, R gene and CED-4.

catalytic domain (*Tanoue et al., 2000*). Specifically, the threonine or tyrosine in the TXY motif as the activation loop plays pivotal roles in the signal transduction pathway. Remarkably, the average abundance of threonine or tyrosine (TXY), the most important enzymes in the *B. distachyon* inbred lines, were 4.67 and 3.91, respectively, and 4.62 and 3.81 in Bd21, respectively (Table S7), suggesting that these amino acids were relatively constant. We also found that the conserved domains of eleven MPKs reported in comprehensive plant species were highly conserved in individual MPKs of *B. distachyon* inbred lines (Figs. S8 and S9). These analyses revealed that the activation loop TEY in groups A, B, and C, and TDY in group D, were consistent with results from previous studies. Most notably, the activation loop TEY motif in all *B. distachyon* MPK11s was replaced by the MEY motif (Fig. 1). MAPKs harbored a CD domain featuring a cluster of negatively-charged amino acids with consensus sequences M/L-L-A/V-F-D-P-X2-R-P/I-T/S-A/V-X-E-A-L (Fig. 1) that bind the basic residues at the N-terminus of the docking site in MAPK-interaction proteins (*Jiang et al., 2018*). MPK7-2s belonged to leucine-rich

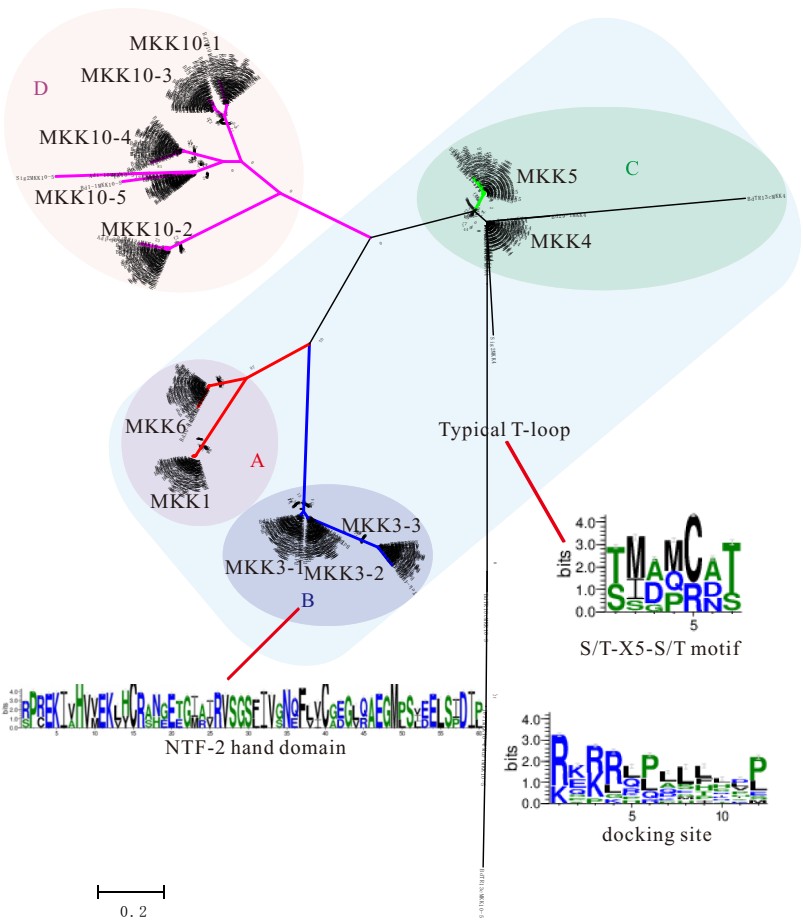

**Figure 2** **Phylogenetic distribution of 618 MKKs among diverse *B. distachyon* inbred lines.** MKKs are divided into four clades (A–D). Sequence features show in the form of web logos representing the NTF-2 domain conserved in the C-terminal extensions of MKK3s. Web logos analysis shows amino acid distribution of conserved S/T-X$_5$-S/T motif in MKKs (groups A–C) and the docking site conserved in N-terminal extension of MKKs.

repeat receptor kinases (LRR-RKs) that had LRR domains and an NB-ARC domain, only appearing in seven kinds of *B. distachyon* inbred lines (Fig. 1). Moreover, the DLK active site within the MPK7-1s signature was conserved; however, Luc1MPK7-1 and ABR6MPK7-1 were replaced with a DLN motif. Foz1MPK7-2 and Mur1MPK7-2 with DLN motifs were treated similarly (Fig. S8). Specifically, all MPK11s had a DLR motif instead of a DLK active site (Fig. S8). Furthermore, an elongation factor hand (EF-hand) calcium binding protein (CBP) with the consensus sequences "NDNNEQHASDQER" was observed in all *B. distachyon* MPK17s at their C-terminal end (Fis. 1, 5A and 5B). Further analysis indicated that 13 MPK17 members (group T+: BdTR13cMPK17, Bis-1MPK17, Bd21-3MPK17, BdMPK17, Adi-2MPK17 and BdTR13aMPK17; group EDF+: Tek-2MPK17, Tek-4MPK17, Bd29-1MPK17, ABR9MPK17, BdTR8iMPK17, BdTR7aMPK17, and Bd1-1MPK17) had a mutation in which E changed to D, which only resulted from a single nucleotide substitution of A changed to C (Fig. 5C). This mutation may be have no effect on the function due to the canonical EF-hand (for example the

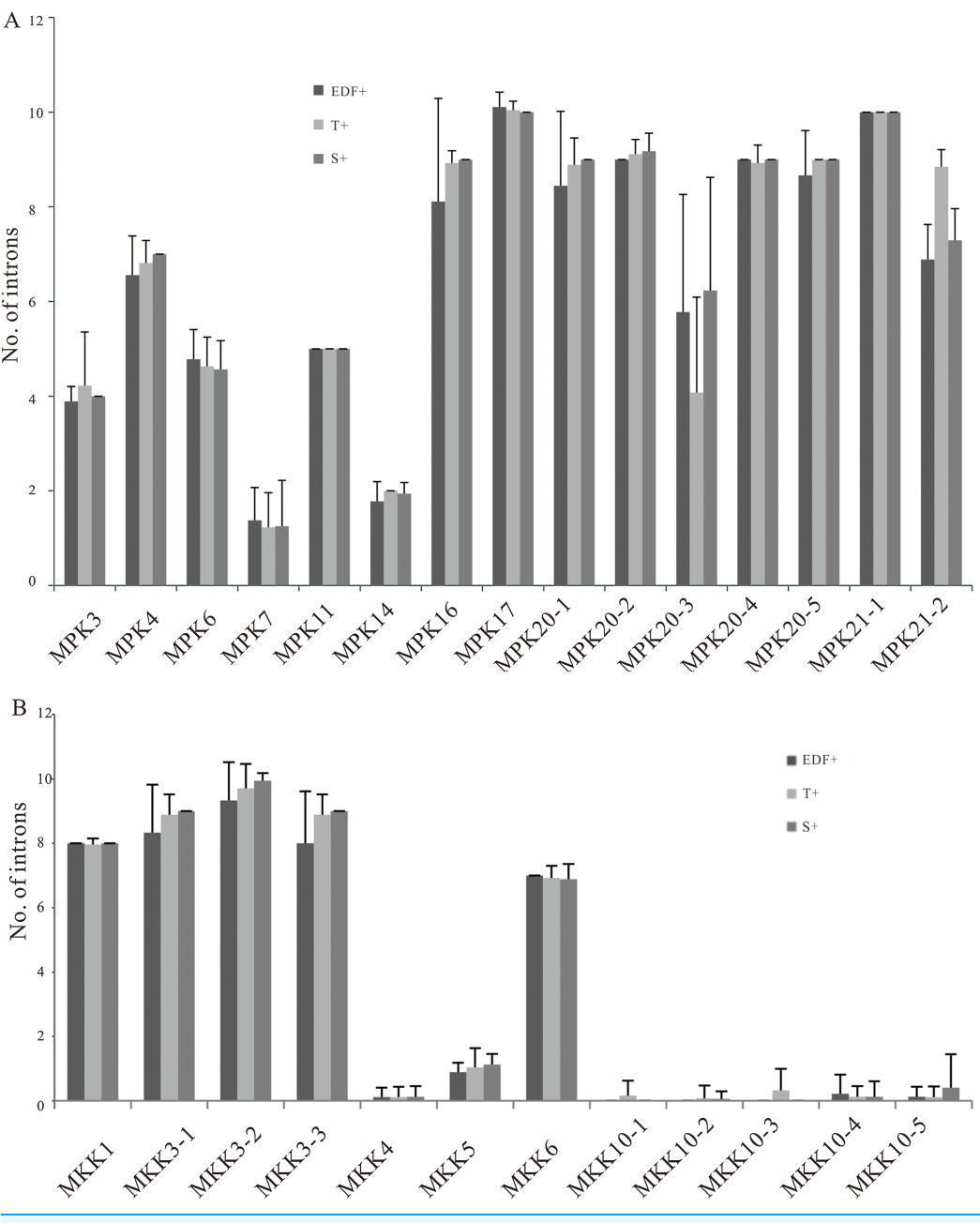

**Figure 3 Intron number polymorphisms of MPKs (A) and MKKs (B) from 53 diverse *B. distachyon* inbred lines.** Different colors represent different genomic groups.

calcineurin B-like (CBL) protein), that are characterized by a conserved Asp (D) and Glu (E) residue with completely constant spacing (*Kolukisaoglu et al., 2004*; *Jiang et al., 2020*). Remarkably, the MPK17s exhibited the conserved domain specific to *B. distachyon* members compared to other plant species, especially eudicots (Fig. 5D).

As same as the MAPKs, MKKs also contained some important domains or motifs including the activation or T-loop S/T-$X_5$-S/T motif (*Asai et al., 2002*), the docking site

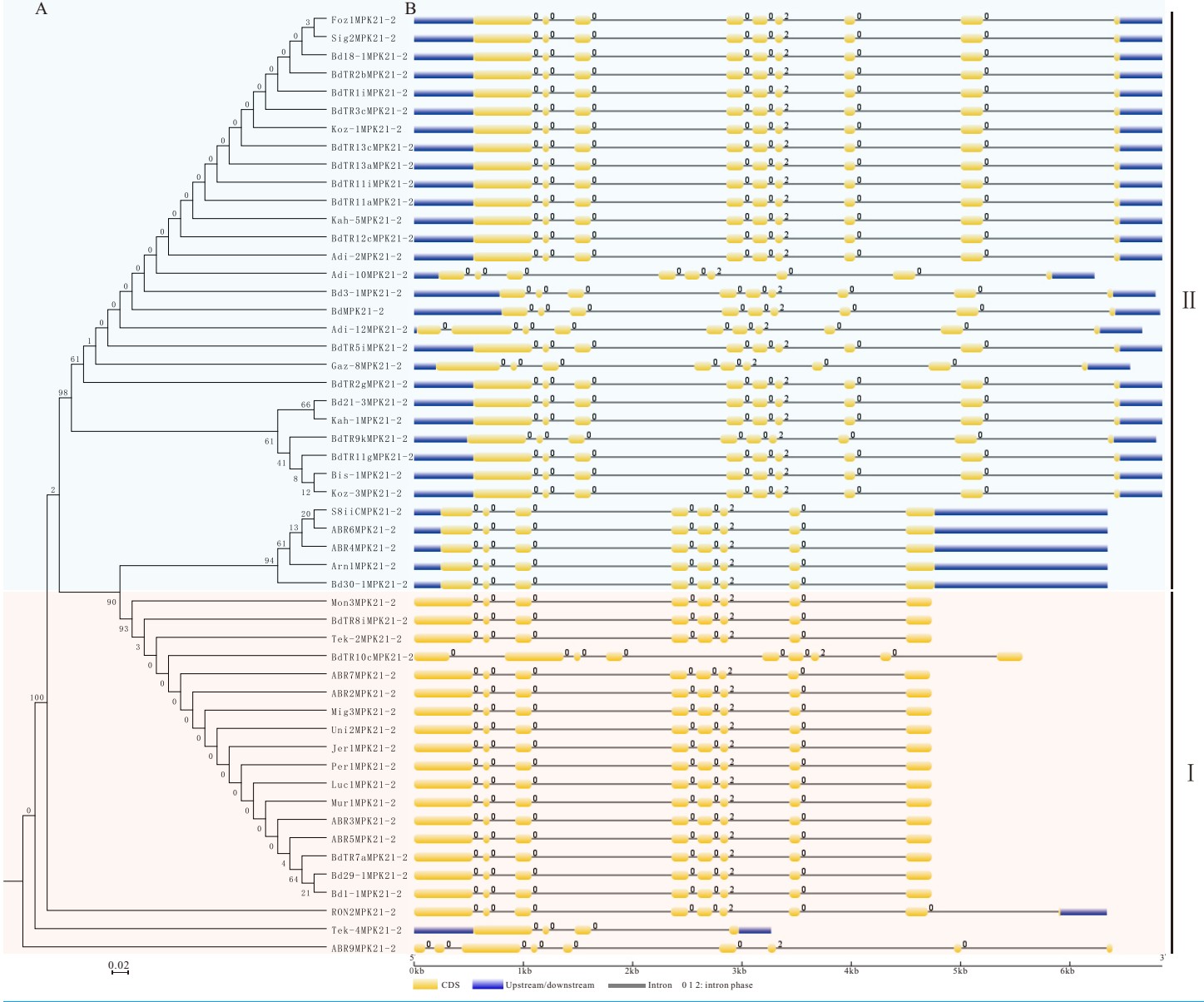

**Figure 4 Gene structure of *MPK21-2* genes.** (A) Maximum Likelihood phylogenetic trees of the full CDS sequences of genes encoding MPK21-2 from diverse *B. distachyon* inbred lines. (B) The exon/intron structure of each *MPK21-2* gene was displayed. Yellow boxes represent exons, gray lines represent introns and blue boxes represent UTRs. The exons are drawn to scale.

(K/R$_{2-3}$X$_{1-5}$L/IXL/I) in the N-terminal domain (*Bardwell & Shah, 2006*; *Jiang & Chu, 2018*), the GxGxxGxV motif in the NB domain and HK-X$_6$-ALK motif in the ATP binding site (*Hadiarto et al., 2006*), and the active site D(I/L/V)K motif (*Goyal et al., 2018*). A detailed analysis of the conserved sequences of MKKs was displayed in alignment of individual MKKs (Figs. S10 and S11). Groups A, B, and C MKKs had the typical T-loop S/T-X$_5$-S/T motif, while group D MKKs (MKK10s) had a part mutation in the phosphorylation site which coincided with a wide range of plant species (Fig. 2; Fig. S10) (*Jiang & Chu, 2018*). Interestingly, the average abundance of serine or threonine

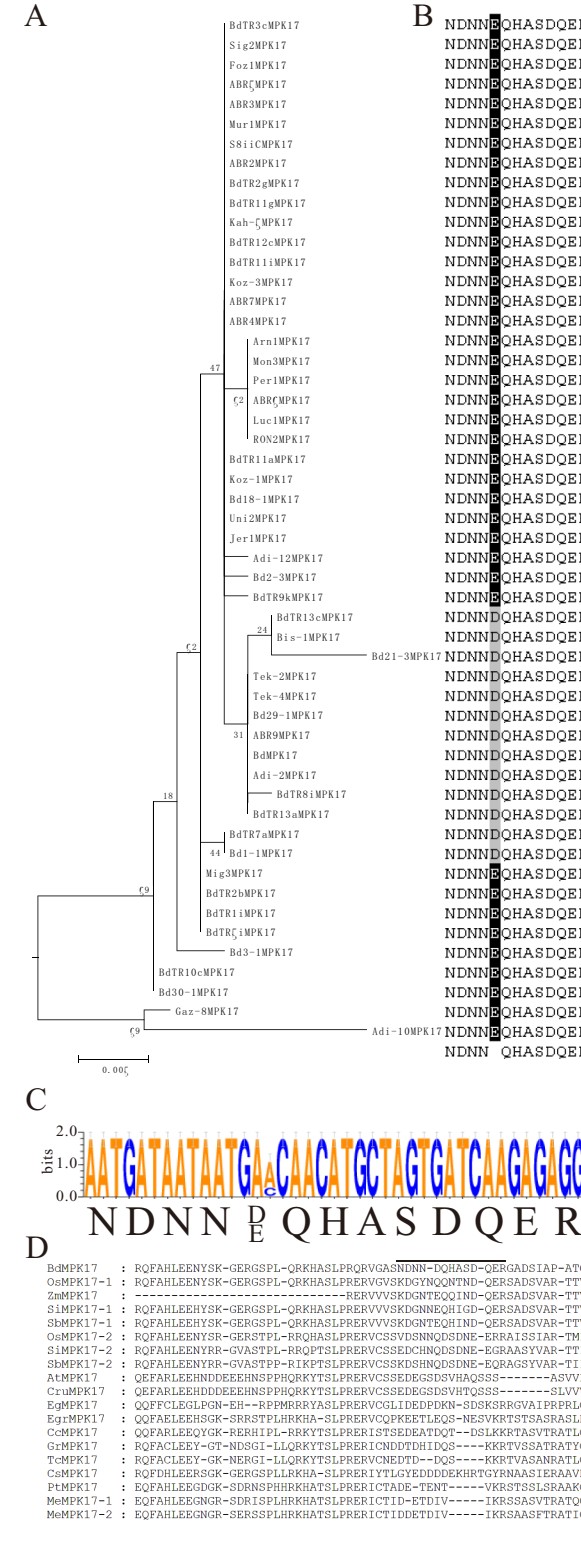

**Figure 5 Conservation and consensus pattern of the EF-hand motif of *B. distachyon* inbred lines MPK17s.** (A) Maximum Likelihood phylogenetic trees of the full sequences of genes encoding MPK17s from *B. distachyon* inbred lines. (B) ClustalW multiple-sequence alignment of the region containing the EF-hand motif within MPK17s. Conserved residues are shown in dark colors. (C) Consensus pattern and sequence logo of the EF-hand motif generated using the Weblogo3

**Figure 5** (continued)

application (http://weblogo.threeplusone.com/). The overall height of each stack indicates the sequence conservation at that position (measured in bits), whereas the height of symbols within the stack reflects the relative frequency of the corresponding base at that position. (D) ClustalW multiple-sequence alignment of the region containing the EF-hand motif within MPK17 in different plant species. The specific domain was marked by thick line. At: *Arabidopsis thaliana*; Bd: *Brachypodium distachyon*; Cru: *Capsella rubella*; Cc: *Citrus clememtina*; Cs: *Cucumis sativus*; Eg: *Erythranthe guttata*; Egr: *Eucalyptus grandis*; Gr: *Gossypium raimondii*; Me: *Manihot esculenta*; Os: *Oryza sativa*; Pt: *Populus trichocarpa*; Sb: *Sorghum bicolor*; Si: *Setaria italica*; Tc: *Theobroma cacao*; Zm: *Zea mays*.

(S/T-X$_5$-S/T), which were the most crucial amino acids in *B. distachyon* inbred lines, were 7.2 and 3.32, respectively, while the same results were 7.2 and 3.32 in Bd21 (Table S7), respectively. This suggests that these amino acids remained constant. We speculated that the MKKs may have experienced fundamental functional conservation. Further analysis showed that variations of some the conserved MKKs were also present. For instance, in addition to BdTR13CMKK4 being replaced by a DIL motif, the D(I/L/V)K active site within the signature of MKKs was conserved despite occasional variations (Fig. S10). We found the HRPTGRCYALK motif in the ATP binding site of MKK5 members, however, BdTR3cMKK5 was replaced by the HRPPGRCYALK motif (Fig. S10). Furthermore, our data showed that the nuclear transport factor 2 (NTF2) domains existed in all MKK3s from *B. distachyon* inbred lines (Fig. 2).

## The differential evolution of *MKK10* paralogs with tandem duplications

Gene duplication was the necessary material source for evolutionary novelty, leading to the gene responsible for the gene families (*Lynch & Conery, 2000*). In addition, some tandem duplication was observed in monocot *MKK10* paralogues, such as in *B. distachyon* (*Jiang & Chu, 2018*). A 6,7-dimethyl-8-ribityllumazine (*DMRL*) synthase gene was observed between two *MKK* members (*Jiang & Chu, 2018*). We surveyed the duplicated genes in different *B. distachyon* inbred lines genome to further comprehend the duplication and evolutionary events of the *B. distachyon MKK10* paralogues. As expected, most of *MKK10* paralogues with the exception of five kinds of *B. distachyon* inbred lines (Mon3, Bd3-1, Adi-10, BdTR10c, and Gaz-8) presented tandem duplication in the canonical form of the *MKK-DMRL-MKK* model with occasional variations (Fig. 6). In addition, Tek-4 had the *PNN* (pinin) gene instead of the *DMRL* gene between two *MKK* gene members (Fig. 6). Twenty *B. distachyon* inbred lines had the same canonical model with Bd21 in the form of the *MKK-DMRL-MKK-MKK* model. Eight *B. distachyon* inbred lines possessed the tandem duplication in the form of the *MKK-DMRL-MKK* model (Fig. 6). Moreover, some small variations were also found in other *B. distachyon* inbred lines. For example, we also found the tandem gene clusters with the *MKK-DMRL-UDPGT-UDPGT-MKK* (*UDPGT*: UDP-glucoronosyl and UDP-glucosyl transferase) model in Mur1, *MKK-DMRL-MKK-ChaC-MKK* (*ChaC*: ChaC-like protein) model in BdTR12c, and *MKK-DMRL-PK-PK-MKK* (*PK*: protein tyrosine kinase) model in Sig2 (Fig. 6). These results indicated that the tandem *MKK10* gene clusters in *B. distachyon*

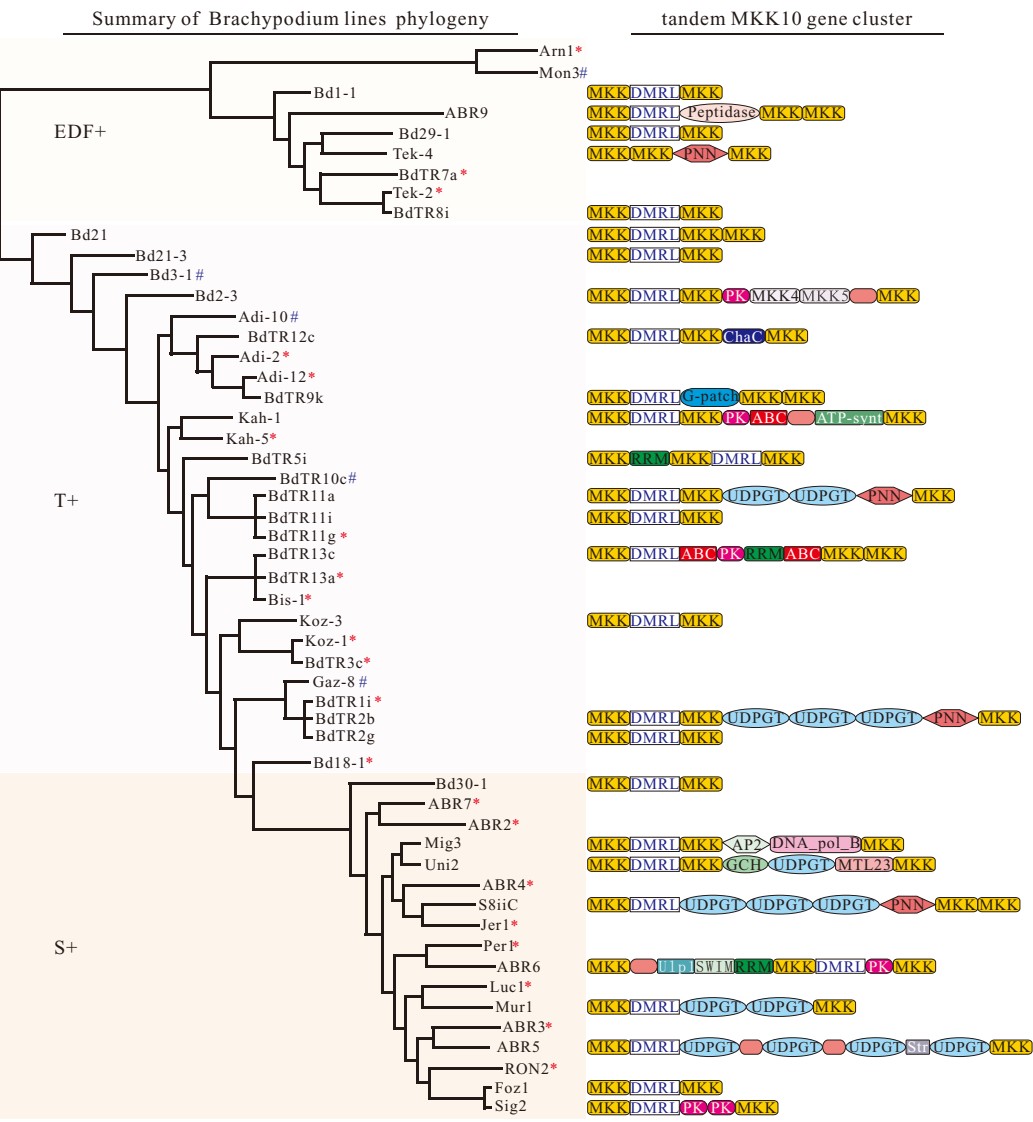

**Figure 6 The diversity and evolution of the fate of an ancestral locus having *B. distachyon* inbred lines *MKK10* genes in tandem position.** Phylogenetic relationships among 53 diverse *B. distachyon* inbred lines were investigated. The phylogenetic tree is modified from BrachyPan (https://brachypan.jgi.doe.gov/). The variants of ancestral tandem *MKK10* gene clusters in *B. distachyon* inbred lines are shown on the right. The red asterisk indicates the gene cluster models of these inbred lines are same as Bd21, while blue pound represents no tandem duplication events. Gene or protein names: MKK (MAPK kinase 10); DMRL ( DMRL synthase ); Peptidase (Peptidase_C48); PNN (pinin); ChaC (ChaC-like protein); G-patch (glycine rich nucleic binding domain); ATP-synt (ATP synthase subunit C); RRM (RNA recognition motif protein); UDPGT (UDP-glucoronosyl and UDP-glucosyl transferase); ABC (ABC transporter); PK (Protein tyrosine kinase); AP2 (AP2/EREBP transcription factor); DNA_pol_B (DNA polymerase family B); GCH (Predicted glycine cleavage system H protein); MTL23 (Methyltransferase-like protein 23); Ulp1 (Ulp1 protease); SWIM (SWIM zinc finger); Str (Strictosidine synthase).

inbred lines originated in the common ancestral genomic contexts and a certain variation developed in order to adapt to the environment differences including light, temperature, or elevation.

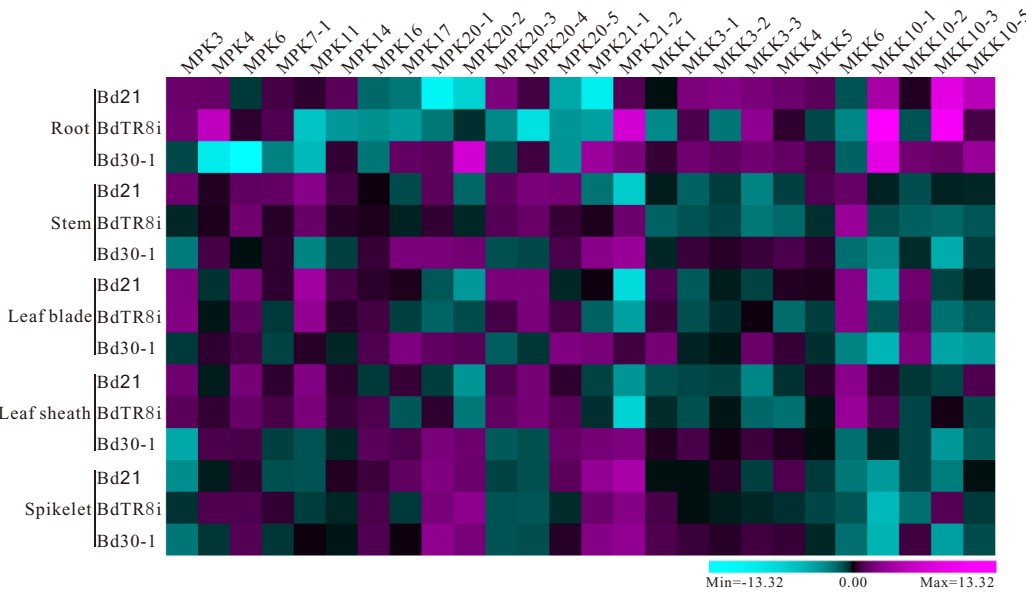

**Figure 7 Quantitative expression variation of three diverse *B. distachyon* inbred lines *MPK* and *MKK* genes in five different tissues.** Root, stem, leaf blade and leaf sheath samples were collected at the 8–9 leaf stage of BdTR8i, Bd21 and Bd30-1 seedling, respectively. While spikelet samples were harvested at *B. distachyon* flowering one to two weeks according to their different flower times.

## Expression variation in the *MPK* and *MKK* gene family in three selected genomes

The expression profiles for five different tissues (root, stem, leaf blade, leaf sheath, and spikelet) and abiotic stresses (salt, drought, and heat) of *MPK* and *MKK* genes were performed (Table S8) to explore the expression variation in Bd21, BdTR8i, and Bd30-1, which belong to three different genetic groups. We observed the different expression levels of 15 *MPK* and 11 *MKK* genes apart from the *MKK10-4* gene (Figs. 7 and 8), which may be involved with the barely detectable low expression level of the *MKK10-4* gene observed in previous research (*Jiang & Chu, 2018*). Among these *MPK* and *MKK* genes, most genes showed distinct quantitative expression patterns in their different genetic backgrounds. For instance, the expression level of *MPK4s* at root was higher in BdTR8i and moderate in Bd21 compared with in Bd30-1 (Fig. 7). *MPK3s* had the same reaction after 6 h of drought treatment (Fig. 8). *MKK10-3s* had higher expression in spikelet's in BdTR8i compared with in Bd21 and Bd30-1 (Fig. 7). Moreover, these genes also had similar expression patterns in different genetic backgrounds, such as *MPK16s*, *MPK20-4s*, *MKK3-2s*, and *MKK10-1s* in spikelet, *MKK5s* in leaf sheath, *MPK6s* and *MKK10-2s* in leaf blade, *MPK16s* and *MKK10-5s* in stem, *MPK16s*, and *MPK20-5s* and *MKK10-1s* in root (Fig. 7). *MKK4s* had a higher expression under drought conditions in BdTR8i compared with in Bd21 and Bd30-1 (Fig. 8). *MPK3s* had a relatively low expression under heat treatment in three backgrounds, while the expressions of *MPK14s* and *MPK20-1s* were opposite (Fig. 8). Expression variations were also observed in the different tissues and/or abiotic stresses. For example, *MKK10-1s* were more highly expressed in three genetic

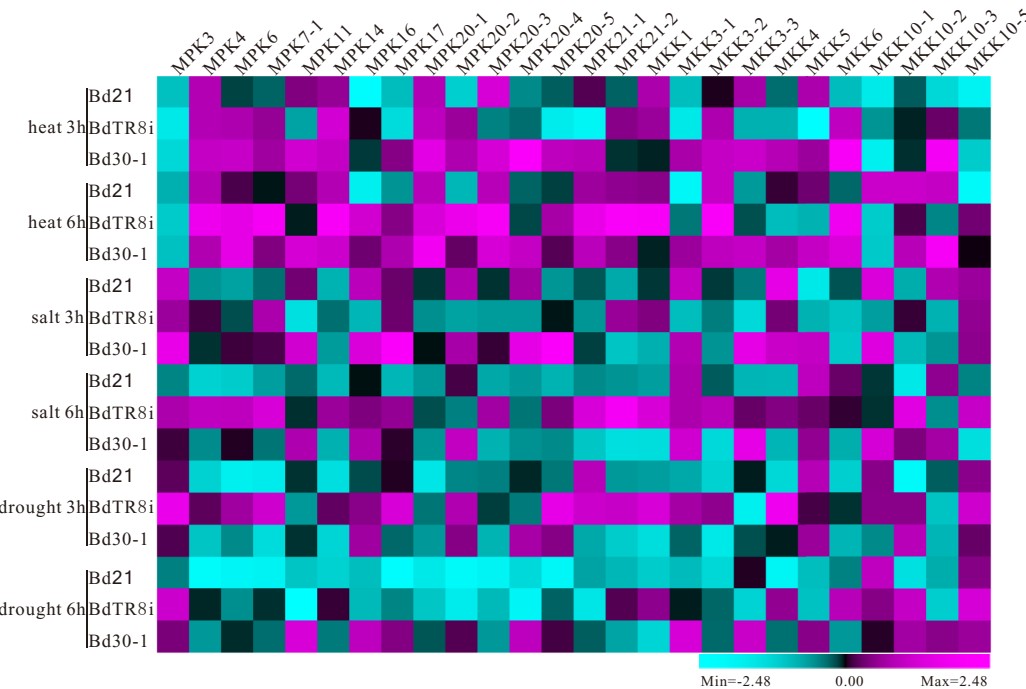

**Figure 8 Expression patterns of *MPK* and *MKK* genes in Bd21, BdTR8i and Bd30-1 seedlings under different abiotic stresses.**

backgrounds in roots than in other tissues (Fig. 7). Almost all genes had higher expression levels after 6 h of exposure to three abiotic stresses in BdTR8i (Fig. 8). We further found that certain genes had unique expression profiles in specific tissues of a particular genetic background. For example, *BdTR8iMPK4* was highly expressed in only the root (Fig. 7) and *BdTR8iMKK1* was highly expressed in heat, salt, and drought stresses (Fig. 8). The striking variations of *MPK* and *MKK* gene family members expressed in different genetic contexts increases the diversity of the potential biological functions of these genes.

# DISCUSSION

## Exon-intron compositions with conservative and divergent patterns

The conservation of exon length was associated with constraints of the gene function of organisms (*Davila-Velderrain, Servin-Marquez & Alvarez-Buylla, 2014*). The non-coding regions, such as the intron, may affect gene functions by a gradual deletion, which may be the result of recombination throughout the evolution of the intron (*Hu, 2006*). Therefore, we investigated the exon-intron composition of the corresponding *MPKs* and *MKKs*. Our results showed that the exon-intron architecture, including lengths and numbers of intron, intron phase, and lengths of UTR, was generally conserved in corresponding orthologs (Figs. S6 and S7). However, some variability was also found. For example, type II *MPK21-2*s harbored 5′-UTR and 3′-UTR which were absent in type I *MPK21-2*s (Fig. 4), although they were in agreement with their phylogenetic relationships (Fig. 1), indicating that they may have a difference in expression and functional divergence. The UTR length-dependent functional specificity significantly increases the coding

capacity of the genome that regulates multiple plant process, including nutrient homeostasis, stress responses, and plant growth and development (*Srivastava et al., 2018*). In addition, there was a large difference in the intron lengths and numbers among *B. distachyon MPK7-1*s and *MPK20-3*s (Fig. S6). A detailed analysis showed that the fourth intron of *BdTR8iMPK20-3* was shorter than *BdMPK20-3* and *Bd30-1MPK20-3* (Fig. S6). Moreover, *BdTR8iMPK20-3* had a lower expression level in roots and spikelets compared to the other corresponding members as described in a previous study that intron lengths were correlated with gene expression (*Rose et al., 2016*). Our findings indicated that the exon-intron composition affect the evolutionary patterns and expression efficiency of *MPK* and *MKK* orthologs.

## Tandem duplications contributed to *MKK10s* gene expansion

Our analysis suggested that *MKK10* paralogs undergo an ancient tandem duplication event with differential evolution. Further examination of tandem *MKK10* gene clusters revealed that a *DMRL* gene often occurred (Fig. 6). These results are supported by previous studies (*Jiang & Chu, 2018*), indicating that they were derived from common ancestral genomic contexts. However, some variations have also been found among *B. distachyon* inbred lines such as an insertion of *RRM* (RNA recognition motif protein) instead of the *DMRL* gene (Fig. 6). This may result in a difference of gene expression. Indeed, tandem duplicates generally show more similar expression patterns than remote duplicates (*Dai, Xiong & Dai, 2014*; *Lan & Pritchard, 2016*) and preferentially retain the cis-PPIs (protein–protein interactions) after WGD (*Makino & McLysaght, 2008*, *2012*). Therefore, ancient tandem duplications of *MKK10s* may have contributed to gene expansion and function conservation and/or divergence during the evolution process of monocots.

## Expression divergence of *MAPKs* and *MKKs* within three *B. distachyon* genetic groups

Tissue-specific expression patterns of *MAPK* and *MKK* genes have been characterized with corresponding functions in plant growth and development. For instance, the expression levels of *AtMKK10* are high in pollen but do not appear in shoot apices, mesophyll cells, or mature leaves (*Yoo et al., 2008*), indicating a potential role in flower tissues (*Jiang & Chu, 2018*). *CaMPK19-2* genes are highly expressed in roots and stems in pepper, while *CaMPK1* is highly expressed in in leaves (*Liu et al., 2015*), which indicates that these genes are expressed preferentially in different tissues and developmental stages (*Wei et al., 2014*). We investigated the tissue-specific expression profiles in different *B. distachyon* inbred lines. The result indicated that most *MPK* and *MKK* genes had quantitative distinct expression patterns among the three different genetic contexts in different tissues and various abiotic stresses. For example, *MPK17* had higher expression levels in the root, stem, leaf blade, and salt treatment in Bd30-1 compared with Bd21 and BdTR8i (Figs. 7 and 8). These results are indicative of the distinct function of *MPK17s*, which may result from the nonsynonymous substitutions at some pivotal amino acid sites in EF-hand CBP motif in their C-terminal extensions (Fig. 5) as described previously (*Yang et al., 2019*). Moreover, *MKK10-3* and *MKK10-5* had similar expression patterns in the leaf blade in

Bd30-1 and BdTR8i and distinct profiles in Bd21 (Fig. 7). These results coincide with the tandem gene cluster model (Fig. 6) and are supported by previous reports that physically linked genes (tandem duplicates) usually had less expression differences than distant genes (*Ghanbarian & Hurst, 2015*; *Lan & Pritchard, 2016*). Furthermore, the *MKK3-2* gene had similar patterns under heat and salt condition (Fig. 8). Taken together, these results suggest that *MAPKs* and *MKKs* had an expression divergence which was correlated with the differential evolution in *B. distachyon* inbred lines.

## CONCLUSION

A total of 799 *MPK* and 618 *MKK* genes were retrieved from 53 kinds of *B. distachyon* inbred lines based on their conserved TXY or S/T-$X_5$-S/T domain, respectively, using bioinformatics approaches. Phylogenetic analyses showed that most MAPKs and MKKs clustered into same branch, with the exception of MPK21-2s, which was divided into two groups, designated as type I and II. Further analysis found that the divergence of MPK21-2 may be involved with the presence of UTRs. *MKK10s* expanded during the evolutionary process by ancient tandem duplications with a differential model. This may have resulted in expression differences and functional divergence. We discovered that the expression of the *MPK* and *MKK* gene members varied in different tissues and across abiotic stresses in three different genetic contexts, suggesting that these genes may have diverse biological functions. Taken together, our results revealed a more comprehensive understanding of the function and evolutionary patterns of MAPKs and MKKs in diverse *B. distachyon* inbred lines.

### Funding
This work was supported by the Shanghai Sailing Project (19YF1414800) chaired by Min Jiang. The funders had no role in study design, data collection and analysis, decision to publish, or preparation of the manuscript.

### Grant Disclosures
The following grant information was disclosed by the authors:
Shanghai Sailing Project: 19YF1414800.

### Competing Interests
The authors declare that they have no competing interests.

### Author Contributions
- Min Jiang conceived and designed the experiments, performed the experiments, analyzed the data, prepared figures and/or tables, authored or reviewed drafts of the paper, and approved the final draft.
- Peng Li performed the experiments, analyzed the data, prepared figures and/or tables, and approved the final draft.

- Wei Wang analyzed the data, prepared figures and/or tables, and approved the final draft.

## Data Availability

The data of RT-qPCR experiments are available in the Supplementary File.

## Supplemental Information

Supplemental information for this article can be found online at http://dx.doi.org/10.7717/peerj.11238#supplemental-information.

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
