# Peer review of "Comparative analysis of MAPK and MKK gene families reveals differential evolutionary patterns in Brachypodium distachyon inbred lines"

_PeerJ, doi:10.7717/peerj.11238_

## Round 0.1 · original submission · Minor Revisions

In addition to addressing each suggestion from the Reviewers, please make sure you address the following concerns:

- Provide additional supplementary files:
----Fasta sequence files of the two gene families
----The protein sequence alignment files
----The gene family trees in the Newick format along with bootstrap values and branch lengths

- The English language and spellings need editing. Copyediting is not provided as a standard publication service. Please ensure the language in this submission is clear and unambiguous, grammatically correct, and conforms to professional standards of courtesy and expression.

Reviewer 1 ·

Basic reporting

Comment 1: Line no. 283-286 says, “To explore the expression variation of the MPK and MKK gene family in Bd21, BdTR8i and Bd30-1 that belongs separately to three different genetic groups, the expression profiles for five different tissues (root, stem, leaf blade, leaf sheath and spikelet) of MPK and MKK genes were preformed”. The authors have performed the expression analysis of all the MKKs and MPKs in the given Brachypodium inbreed lines for five different plant tissues. The MAPK genes are involved in various abiotic stress tolerance responses and developmental processes, which are justified by Goyal et al., 2018 in his paper “Analysis of MAPK and MAPKK gene families in wheat and related Triticeae species”. They have written, “Mitogen-activated protein kinases (MAPKs) are key signaling enzymes involved in the regulation of various aspects of biology in eukaryotic organisms, including cell division, development, metabolism and stress responses”. The authors should consider providing some abiotic stress tolerance data concerning their identified MKK and MPK genes based on the pivotal importance of the MAPK signaling genes in abiotic stress tolerance mechanisms.

Experimental design

Comment 1: Line no. 90-93 says, “To identify these genes in the 53 diverse B. distachyon inbred lines, BLASTP (Altschul et al. 1997) searches were performed by orthologous protein sequences using BdMAPKs and BdMKKs as the query search in BrachyPan (https://brachypan.jgi.doe.gov/) (Gordon et al. 2017)”. The authors have stated the use of orthologous genes for blast-p analysis. A pint-sized explanation can be provided here in this section for doing the same.
Comment 2: Line no. 111-113 says, “Phylogenetic trees based on the alignment of all MAPKs or MKKs were conducted using the maximum likelihood (ML) method with the Jones-Taylor-Thornton (JTT) model, 2000 bootstrap values and partial deletion by the MEGA 6.0 software”. Why consider using 2000 bootstrap replicate values rather than a 1000? An explanation can be provided for the same.
Comment 3: The authors have used the Maximum Likelihood (ML) method to construct the phylogenetic trees and have represented the trees across figures 1 and 2. The authors can improvise on the representation of the trees. The huge number of genes for both the MKKs and MPKs have created a pile of gene ids in a confined space, which is almost impossible to figure out. The authors can also consider the use of Neighbor Joining (NJ) algorithm to represent their phylogenetic trees and provide them in supplemental files.
Comment 4: Line no. 261-263 says, “Gene duplication was considered as the primary force of the species evolution, which resulted in the gene to form the gene families. In addition, some tandem duplication was observed in monocots MKK10 paralogues, such as in B. distachyon (Jiang and Chu 2018)”. The authors should provide a little more clarity in the above-quoted sentence.

Validity of the findings

No comment

Additional comments

Dear authors, firstly I would like to appreciate the hard work that has been put together to compose this manuscript. To better understand the evolution and function of MAPKs and MKKs in B. distachyon inbred lines, the authors conducted their systematical molecular evolutionary analysis among the 799 MAPKs and 618 MKKs, which were identified from 53 B. distachyon inbred lines, respectively. Phylogenetic analyses and intron-exon patterns were analyzed followed by the expression analysis of all the MKKs and MPKs. These findings can provide new insights into the functional evolution of genes in closely inbred lines. Nevertheless as a reviewer of the scientific community, These are some of the aspects of the manuscript, which can be touched upon and improvised.

·

Basic reporting

No Comment

Experimental design

No Comment

Validity of the findings

No Comment

Additional comments

The manuscript entitled "Comparative analysis of MAPK and MKK gene families reveals
differential evolutionary patterns in Brachypodium distachyon
inbred lines" submit by Jiang et al. is a robust analysis of MAPK and MKK gene families in 53 inbred lines of Brachypodium distachyon. The analysis is good and presentation of the entire manuscript is also impressive. The authors have identified a total of 799 MAPKs and 618 MKKs in 53 inbred lines of B. distachyan, respectively. Theses MAPKs and MKKs are divided into four groups as reported earlier in Arabidopsis and other plants. They also analysed the expression of these genes in different tissues. The highlight of the study is revealing the role UTRs on structure and evolution of MPK21-2 genes and differential evolution of MKK10 paralogues with ancient tandem duplication. I overall liked the manuscript, however I have following very minor concerns that authors may choose to attend:
1. I would prefer to highlight the specific numbers of MAPKs(14/15/16) and MKKs(12) members in each lines of B. distachyon in the abstract.
2. Since authors have performed MSA of 53 different inbred lines of B. distachyon, I was looking for a conserved domain specific to B. distachyon members compared to other plant species. If possible, this information may be added in Figure 5.
3. Authors may provide a bit more details about the age of plants used from which tissues were harvested for expression analysis in the legend of Figure 7.

---

## Round 0.2 · Minor Revisions

Your corrections and updates look great. Please make sure you provide the consistent sequence IDs between the tables S1 and S5 trees and elsewhere. ON the NWK tree files, suffix the existing names with a "_gene_ID" for easy recognition of the gene and the data in the respective file. Also in table S1 add the best match corresponding gene ID of the B. distachyon BD21 reference genome v3. This will help readers evaluate your data and improve the annotation of the BD21 reference genome.

·

Basic reporting

No comment

Experimental design

No comment

Validity of the findings

No comment

Additional comments

The Authors have appropriately modified the revised version of the manuscript. I have no further question for this article.

---

## Round 0.3 · Minor Revisions

Thank you so much for sending us all the updates in a timely manner. The English language still needs corrections. Please hire a professional to help you tidy up the manuscript. The journal PeerJ does not provide English Language editing as part of the review process.

---

## Round 0.4 · accepted · Accept

Please make sure all the weblinks in the manuscript are working or at least convert them into citations. For example, the link http://gsds.cbi.pku.edu.cn/ is not working. Check if it is a firewall issue.